# Low-Density Lipoprotein Cholesterol, Structural Atherosclerosis, and Functional Atherosclerosis in Older Japanese

**DOI:** 10.3390/nu15010183

**Published:** 2022-12-30

**Authors:** Yuji Shimizu, Hirotomo Yamanashi, Yukiko Honda, Fumiaki Nonaka, Jun Miyata, Shin-Ya Kawashiri, Yuko Noguchi, Seiko Nakamichi, Yasuhiro Nagata, Takahiro Maeda

**Affiliations:** 1Department of General Medicine, Nagasaki University Graduate School of Biomedical Sciences, Nagasaki 852-8501, Japan; 2Department of Cardiovascular Disease Prevention, Osaka Center for Cancer and Cardiovascular Diseases Prevention, Osaka 536-0025, Japan; 3Leading Medical Research Core Unit, Nagasaki University Graduate School of Biomedical Sciences, Nagasaki 852-8523, Japan; 4Department of Community Medicine, Nagasaki University Graduate School of Biomedical Sciences, Nagasaki 852-8523, Japan; 5Department of Islands and Community Medicine, Nagasaki University Graduate School of Biomedical Sciences, Nagasaki 853-0031, Japan; 6Nagasaki University Health Center, Nagasaki 852-8521, Japan

**Keywords:** LDL, structural atherosclerosis, functional atherosclerosis, CIMT, CAVI, older, endothelial repair

## Abstract

Aggressive endothelial repair results in the progression of both structural and functional atherosclerosis, while insufficient endothelial repair worsens functional but not structural atherosclerosis. Aging increases the risk of inadequate endothelial repair. Since low-density lipoprotein cholesterol (LDLc) activates endothelial repair, LDLc may be positively associated with structural atherosclerosis but inversely associated with functional atherosclerosis in older individuals. This cross-sectional study analyzed 1458 participants aged 60 to 79 years. We defined structural atherosclerosis as a carotid intima-media thickness (CIMT) of at least 1.1 mm and functional atherosclerosis as a cardio-ankle vascular index (CAVI) of at least 9.0. LDLc was significantly positively associated with structural atherosclerosis and significantly inversely associated with functional atherosclerosis, independently of known cardiovascular risk factors. For 1 standard increment of LDLc (28 mg/dL for men and 29 mg/dL for women), the odds ratios and 95% confidence intervals after adjustment for known cardiovascular risk factors were 1.28 (1.10, 1.50) for structural atherosclerosis and 0.85 (0.75, 0.96) for functional atherosclerosis. LDLc activates endothelial repair, which results in the development of structural atherosclerosis but maintains endothelial function in older individuals. To evaluate atherosclerosis in clinical practice, the combination of structural and functional assessment of atherosclerosis could be informative.

## 1. Introduction

Endothelial repair plays an important role in atherosclerosis progression, resulting in increased arterial stiffness. Atherosclerosis can be categorized into two primary types: structural, which is evaluated by the carotid intima-media thickness (CIMT), and functional, which is evaluated by the cardio-ankle vascular index (CAVI). Aggressive endothelial repair exacerbates both types, while insufficient endothelial repair worsens only the functional type [1].

Because aging is associated with an increased demand for endothelial repair but also decreased endothelial repair activity [2], the impact of insufficient endothelial repair is more severe in older individuals. Therefore, the main cause of functional atherosclerosis progression in this population may be insufficient endothelial repair.

Therefore, in older individuals, aggressive endothelial repair results in worsened structural atherosclerosis [1], while insufficient endothelial repair is a primary contributor to the progression of functional atherosclerosis.

Low-density lipoprotein cholesterol (LDLc) directly contributes to the development of structural atherosclerosis [3] by activating inflammation [4], even when levels are within the normal range [5]. Structural atherosclerosis requires CD34-positive cells [1], which contribute to endothelial repair [6], while a shortage of CD34-positive cells results in functional atherosclerosis but not structural atherosclerosis [1].

Since LDLc was also reported to increase the proliferation of CD34-positive cells [7], individuals with high LDLc levels should have sufficient circulating CD34-positive cells to induce endothelial repair.

In older people, therefore, LDLc may be positively associated with structural atherosclerosis and inversely associated with functional atherosclerosis as a result of activating endothelial repair.

To test these hypotheses, we conducted a cross-sectional study in older individuals.

## 2. Materials and Methods

### 2.1. Study Population

We performed health checkups to evaluate atherosclerosis in Goto city, in the western part of Japan, over a period of 3 years. A detailed description of this survey was published previously [8].

The participants in the present study were those who underwent health checkups between 2017 and 2019 and comprised 2261 individuals (850 men and 1411 women) aged between 60 to 79 years. Of these, 803 individuals who had no CAVI data were excluded. The remaining participants, comprising 1458 elderly Japanese individuals (513 men and 945 women) with a mean age of 70.1 (standard deviation [SD] 5.4), were enrolled in the study.

Which arterial stiffness markers were measured depended on the size of each examination facility. At larger facilities, both the CIMT and CAVI were assessed, while at smaller facilities only the CIMT was evaluated.

Due to insufficient staff, it was not possible to survey the entire city population over the course of 1 year. Therefore, we conducted the survey in sequential areas of Goto city and surveyed the entire population over 3 years.

Written informed consent was obtained from all participants. The study was approved by the Ethics Committee of Nagasaki University Graduate School of Biomedical Sciences (project registration number: 14051404-13).

All procedures in this study were performed in accordance with the ethical standards of the institutional research committee and the 1964 Declaration of Helsinki and its later amendments.

### 2.2. Data Collection and Laboratory Measurements

For each individual, the medical history was ascertained by specially trained interviewers. An automatic body composition analyzer (BF-220; Tanita, Tokyo, Japan) was used to calculate body mass index (BMI, kg/m^2^) after measuring height and weight. Overweight was defined as BMI ≥ 25.0 kg/m^2^ and underweight was defined as BMI < 18.5 kg/m^2^.

After at least 5 min of rest, blood pressure was measured in the sitting position using a blood pressure device (HEM-907; Omron, Kyoto, Japan). Hypertension was defined as systolic blood pressure ≥ 140 mmHg, diastolic blood pressure ≥ 90 mmHg, or use of anti-hypertensive medication.

Fasting blood samples were collected in an EDTA-2K tube and a siliconized tube. High-density lipoprotein cholesterol (HDLc) and LDLc were measured using the direct method, and triglycerides were measured using the enzymatic method, in all cases at SRL, Inc. (Tokyo, Japan). Low HDL was defined as <40 mg/dL while high triglycerides were defined as ≥150 mg/dL. HbA1c was also measured using a standard procedure at SRL, Inc., and diabetes was defined as HbA1c ≥ 6.5% or use of glucose lowering medication.

### 2.3. Evaluation of Structural Atherosclerosis

An experienced vascular examiner measured the CIMT of the left and right common carotid arteries using a LOGIQ Book XP with a 10-MHz ultrasound transducer (GE Healthcare, Milwaukee, WI, USA). The maximum values were calculated using semi-automated digital edge-detection software (IntimaScope; MediaCross, Tokyo, Japan) and a previously described protocol [9]. IntimaScope software was used to increase the accuracy and reproducibility of CIMT measurements. This software semi-automatically recognizes the edges of the internal and external membranes of each artery and automatically determines distances at a sub-pixel level, estimated to be 0.01 mm [10]. Structural atherosclerosis was defined as CIMT ≥ 1.1 mm, as in previous studies [11,12].

### 2.4. Evaluation of Functional Atherosclerosis

Brachial-ankle pulse wave velocity (PWV) measurements are generally used to evaluate functional arterial stiffness. Since PWV measurements can be strongly affected by blood pressure [13], the CAVI was recently developed in Japan to avoid the confounding effects of blood pressure [14]. In the current study, the CAVI was determined using a VaSera VS-1000 vascular screening system (Fukuda Denshi, Tokyo, Japan), with the participant resting in a supine position.

According to the manufacturer’s recommendations, CAVI values were categorized as follows: normal, <8.0; borderline, ≥8.0 to <9.0; and abnormal, ≥9.0. These categories were also used in previous studies [15,16]. The borderline category corresponds to the “early atherosclerosis” category in our previous study [17]. In the current study, therefore, as in our prior report, we defined atherosclerosis based on CAVI values as follows: normal, <8.0; early atherosclerosis, ≥8.0 to <9.0; and atherosclerosis, ≥9.0.

### 2.5. Statistical Analysis

The clinical characteristics of the study participants were compared by LDLc tertiles, which were categorized by sex-specific values. The respective LDLc tertiles for men and women were <104 and <113 mg/dL for T1 (low), 104–125 and 113–137 mg/dL for T2 (medium), and ≥126 and ≥138 mg/dL for T3 (high).

Data are presented as means ± SD or percentages (%). Trend tests were also used in a general linear regression analysis to compare baseline variables according to LDLc level.

Logistic regression was used to calculate odds ratios (ORs) and 95% confidence intervals (CIs). These were used to determine the associations between structural and functional atherosclerosis (ORs) and between LDLc levels and atherosclerosis (both structural and functional) (CIs).

Three different models were used to adjust for confounding factors. In Model 1, we adjusted only for sex and age (years). In Model 2, we further adjusted for weight (underweight, normal, overweight), hypertension (no, yes), drinking status (none, often, daily), smoking status (never, former, current), high triglycerides (no, yes), low HDLc (no, yes), and diabetes (no, yes). In Model 3, in addition to the variables included in Model 2, we adjusted for lipid-lowering medication use (no, yes). These associations were all validated by goodness of fit using the Hosmer–Lemeshow test.

Finally, we performed sex-stratified sensitivity analyses to estimate the associations between LDLc levels and both structural and functional atherosclerosis.

All statistical analyses were performed using the SAS system for Windows (version 9.4; SAS Inc., Cary, NC, USA). A *p*-value < 0.05 was considered statistically significant.

## 3. Results

Of the participants, 237 were diagnosed with structural atherosclerosis and 494 with functional atherosclerosis.

### 3.1. Characteristics of the Study Population

Table 1 shows the characteristics of the study population in relation to LDLc, which was inversely associated with age, daily alcohol consumption, and lipid-lowering medication use.

### 3.2. Association between Structural and Functional Atherosclerosis

The associations between structural atherosclerosis and each category of functional atherosclerosis (normal, early atherosclerosis, and atherosclerosis) are shown in Table 2. There was a significant positive association between structural and functional atherosclerosis in the sex- and age-adjusted model (Model 1), but this association disappeared in the multivariable models (Models 2 and 3).

### 3.3. Association between LDLc and Both Structural and Functional Atherosclerosis

Table 3 shows the associations between LDLc and both structural and functional atherosclerosis. LDLc was significantly positively associated with structural atherosclerosis but significantly inversely associated with functional atherosclerosis. These associations were consistent even after further adjustment for known cardiovascular risk factors.

### 3.4. Sex-Specific Associations between LDLc and Both Structural and Functional Atherosclerosis

For sensitivity analysis, we repeated the main analysis with stratification by sex and found essentially the same associations. The age-adjusted ORs (95% CIs) of structural and functional atherosclerosis per 1-SD increment of LDLc were 1.41 (1.14, 1.74) and 0.77 (0.64, 0.93) for men (n = 513), respectively, and 1.07 (0.87, 1.32) and 0.85 (0.73, 0.996) for women (n = 945), respectively.

### 3.5. Association between LDLc and Functional Atherosclerosis According to the Status of Structural Atherosclerosis

In an additional analysis, we repeated the main analysis with stratification by structural atherosclerosis and found a significant inverse association between LDLc and functional atherosclerosis but only in participants without structural atherosclerosis. The sex- and age-adjusted ORs (95% CIs) of functional atherosclerosis per 1-SD increment of LDLc were 0.99 (0.74, 1.33) for those with structural atherosclerosis (n = 237) and 0.78 (0.69, 0.89) for those without it (n = 1221), respectively.

### 3.6. Associations between LDLc and Both Structural and Functional Atherosclerosis by Age Group

We also repeated the main analysis with stratification by age group and found essentially the same associations. The age-adjusted ORs (95% CIs) of structural and functional atherosclerosis per 1-SD increment of LDLc were 1.35 (1.05, 1.74) and 0.83 (0.70, 0.997) for ages 60 to 69 years (n = 754), respectively, and 1.15 (0.97, 1.37) and 0.80 (0.69, 0.93) for ages 70 to 79 years (n = 704), respectively.

## 4. Discussion

The main findings of present study in older individuals were that LDLc was significantly positively associated with structural atherosclerosis but significantly inversely associated with functional atherosclerosis, in both cases independently of known cardiovascular risk factors.

A previous population-based study of 1779 participants without conventional cardiovascular risk factors reported a positive association between LDLc and structural atherosclerosis defined by either vascular ultrasound or coronary artery calcification scores based on computed tomography findings [5]. Another case-controlled, cross-sectional study in 138 participants aged 20 to 79 years who had no cardiovascular risk factors revealed that the main parameters determining the CIMT were age, male sex, systolic blood pressure, and LDLc [18]. Therefore, our results showing a significant association between LDLc and structural atherosclerosis are compatible with those of previous studies. We also found that LDLc was inversely associated with functional atherosclerosis in older participants. However, the mechanism of these associations has not yet been clarified.

A previous population-based, cross-sectional study in 1014 Japanese aged over 40 years showed a significant positive correlation between the CIMT and CAVI, and that this correlation was independent of known cardiovascular risk factors [19]. In the present study, we found a significant positive association between structural atherosclerosis and each type of functional atherosclerosis (normal, early atherosclerosis, and atherosclerosis) in the sex- and age-adjusted model. However, after adjustment for known cardiovascular risk factors, the significant association disappeared. Adjustment for these risk factors may have increased the influence of insufficient endothelial repair for the following reasons.

In conjunction with activated platelets, CD34-positive cells play an important role in endothelial repair, including furthering the progression of structural atherosclerosis. Upon vascular injury, platelets become activated and contribute significantly to vascular homeostasis [20]. Activated platelets induce the proliferation of CD34-positive cells [21] and the differentiation of these cells into megakaryocytes (which are responsible for platelet synthesis) [22]. Platelets also induce differentiation of CD34-positive cells into endothelial cells [23], macrophages, and foam cells [21]. In conjunction with LDLc, these cells contribute to the progression of structural atherosclerosis [7,24].

Our previous study in men aged 60 to 69 years revealed a positive correlation between the CIMT and CAVI but only among participants who had a sufficient number of CD34-positive cells to result in aggressive endothelial repair [1]. Therefore, endothelial repair activity as evaluated by the number of circulating CD34-positive cells should determine the correlation between the CIMT and CAVI.

Bone marrow-derived CD34-positive cells were reported to play an important role in vascular repair [6]. Aging is associated with a decline in bone marrow activity [25,26]. Since the target population of this study was older individuals, the analysis might emphasize the influence of insufficient endothelial repair due to a shortage of CD34-positive cells. Further, insufficient endothelial repair increases the CAVI (functional atherosclerosis) but not the CIMT (structural atherosclerosis).

An independent, significant inverse association between circulating hematopoietic stem cells (CD34-positive cells) and the CAVI was previously reported among older participants in whom the number of circulating CD34-positive cells was below the median value [1]. Since LDLc was also reported to affect the proliferation of CD34-positive cells [7], individuals with high LDLc levels should have sufficient circulating CD34-positive cells to induce endothelial repair. Therefore, LDLc may act as a marker of endothelial repair activity, while functional atherosclerosis may serve as a marker of insufficient endothelial repair in older individuals. Analyzing older individuals may emphasize the influence of insufficient endothelial repair, accounting for the observed inverse association between LDLc and functional atherosclerosis. An additional analysis that evaluated the associations between LDLc and both structural and functional atherosclerosis stratified by age group (60 to 69 years and 70 to 79 years) showed essentially the same results. To evaluate the influence of age on the present associations, further research involving younger individuals is necessary.

Aggressive endothelial repair, which is related to the development of structural atherosclerosis, also decreases the number of circulating CD34-positive cells due to consumption [11,27]. This is consistent with the previous finding of a significant inverse association between baseline atherosclerosis and active arterial wall thickening as evaluated by yearly progression of CIMT [11]. In people with structural atherosclerosis, functional atherosclerosis may develop both due to aggressive endothelial repair and to insufficient endothelial repair caused by a shortage of CD34-positive cells due to consumption. Additionally, in individuals without structural atherosclerosis, functional atherosclerosis may arise only as the result of insufficient endothelial repair. These mechanisms suggest that the presence or absence of structural atherosclerosis confounds the association between LDLc and functional atherosclerosis. This could explain why the additional analysis in this study revealed that only individuals without structural atherosclerosis showed a significant inverse association between LDLc and functional atherosclerosis.

A previous longitudinal study in older individuals aged 67–75 years at baseline paradoxically found that higher LDLc was associated with greater longevity [28]. Another study reported an inverse association between LDLc and atrial fibrillation [29]. Since an increased CAVI may represent a major modifiable risk factor for the development of atrial fibrillation [30], the present study partly explains the potential mechanism of these previous associations: LDLc activates endothelial repair, which results in the development of structural atherosclerosis but maintains endothelial function in older individuals.

The present study has important clinical implications. In general clinical practice, structural and functional atherosclerosis are considered to be essentially the same medical condition. However, from the perspective of endothelial repair activity, there are meaningful differences between them. Therefore, to evaluate atherosclerosis in clinical practice, a combination of structural and functional measures could be informative.

Potential limitations of the present study warrant consideration. CD34-positive cells might have played important roles in our findings, but we have no data on the number of these cells in each individual. Further investigation that includes CD34-positive cell counts is necessary. In addition, this was a cross-sectional study and was therefore unable to establish causal relationships.

## 5. Conclusions

In conclusion, LDLc was positively associated with structural atherosclerosis and inversely associated with functional atherosclerosis in older participants, independently of known cardiovascular risk factors.

## Figures and Tables

**Table 1 nutrients-15-00183-t001:** Characteristics of study population in relation to LDL cholesterol.

	LDL Cholesterol			*p* for Trend
	T1 (Low)	T2	T3 (High)	
No. of participants	486	489	483	
Men, %	35.0	36.2	34.4	0.832
Age	70.8 ± 5.2	69.8 ± 5.3	69.7 ± 5.5	0.003
Overweight, %,	24.5	23.5	25.1	0.853
Underweight, %	7.8	8.6	7.5	0.800
Hypertension, %	67.7	59.9	58.4	0.006
Drinking status: daily, %	18.7	18.2	13.0	0.032
Drinking status, often, %	11.5	13.9	17.0	0.051
Current smoker, %	9.3	9.0	7.2	0.475
Former smoker, %	22.8	20.7	22.8	0.646
High triglycerides, %	12.6	14.1	16.8	0.169
Low HDLc, %	4.7	4.3	2.9	0.313
Diabetes, %	10.1	5.7	8.7	0.040
Lipid-lowering medication use, %	44.2	16.4	10.4	<0.001
CIMT, mm	0.91 ± 0.19	0.90 ± 0.19	0.94 ± 0.20	0.017
CAVI	8.7 ± 1.1	8.5 ± 1.0	8.5 ± 1.1	0.014

TG: triglycerides; LDL: low-density lipoprotein; HDLc: high-density lipoprotein cholesterol; CIMT: carotid intima-media thickness; CAVI: cardio-ankle vascular index. Values are means ± SD. Tertile values of LDL-cholesterol for men were <104 mg/dL for T1 (low), 104–125 mg/dL for T2 (medium), and ≥126 mg/dL for T3 (high). The corresponding values for women were <113, 113–137, and ≥138 mg/dL.

**Table 2 nutrients-15-00183-t002:** Association between structural atherosclerosis and categories of functional atherosclerosis.

	Functional Atherosclerosis			*p* for Trend
	Normal (CAVI < 8.0)	Early Atherosclerosis (CAVI: 8.0–8.9)	Atherosclerosis (CAVI ≥ 9.0)	
**Structural atherosclerosis**				
No. at risk	408	556	494	
No. of cases (%)	37 (9.1)	93 (16.7)	107 (21.7)	
Model 1	Ref	1.58 (1.04, 2.40)	1.62 (1.05, 2.48)	0.048
Model 2	Ref	1.50 (0.99, 2.30)	1.39 (0.90, 2.16)	0.069
Model 3	Ref	1.51 (0.99, 2.30)	1.39 (0.90, 2.16)	0.231

Ref: reference. Model 2: adjusted for sex, age, weight (underweight, normal weight, overweight), hypertension, drinking status (none, often, daily), smoker (never, former, current), high triglycerides, low HDLc, and diabetes. Model 3: adjusted for variables in Model 2 and lipid-lowering medication use.

**Table 3 nutrients-15-00183-t003:** Associations between low-density lipoprotein cholesterol and both structural and functional atherosclerosis.

	LDL Cholesterol (LDLc)			*p* for Trend	1 SD Increment of LDLc
	T1 (Low)	T2	T3 (High)		
**Structural atherosclerosis**					
No. at risk	486	489	483		
No. of cases (%)	70 (14.4)	80 (16.4)	87 (18.0)		
Model 1	Ref	1.28 (0.89, 1.83)	1.48 (1.04, 2.11)	0.031	1.22 (1.06, 1.42)
Model 2	Ref	1.37 (0.95, 1.97)	1.59 (1.10, 2.29)	0.013	1.25 (1.08, 1.45)
Model 3	Ref	1.44 (0.98, 2.10)	1.69 (1.15, 2.50)	0.008	1.28 (1.10, 1.50)
**Functional atherosclerosis**					
No. at risk	486	489	483		
No. of cases (%)	199 (40.9)	151 (30.9)	144 (29.8)		
Model 1	Ref	0.68 (0.51, 0.90)	0.66 (0.50, 0.87)	0.003	0.82 (0.73, 0.92)
Model 2	Ref	0.73 (0.55, 0.97)	0.72 (0.54, 0.96)	0.024	0.85 (0.75, 0.95)
Model 3	Ref	0.73 (0.55, 0.99)	0.73 (0.53, 0.99)	0.040	0.85 (0.75, 0.96)

Ref: reference. SD: standard deviation. Model 1: adjusted only for sex and age. Model 2: adjusted for sex, age, weight (underweight, normal weight, overweight), hypertension, drinking status (none, often, daily), smoker (never, former, current), high triglycerides, low high-density lipoprotein cholesterol (HDLc), and diabetes. Model 3: adjusted for variables in Model 2 and lipid-lowering medication use. Tertile values of LDLc for men were <104 mg/dL for T1 (low), 104–125 mg/dL for T2 (medium), and ≥126 mg/dL for T3 (high). The corresponding values for women were <113, 113–137, and ≥138 mg/dL.

## Data Availability

According to ethical guidelines in Japan, we cannot provide individual data due to participant privacy considerations. In addition, the informed consent obtained does not include a provision for publicly sharing data. Qualified researchers may apply to access a minimal dataset by contacting Prof. Takahiro Maeda, Principal Investigator, Department of General Medicine, Nagasaki University, Nagasaki, Japan at tamaeda@nagasaki-u.ac.jp or the Office of Data Management at ritouken@vc.fctv-net.jp. Information about data requests is also available online at: https://www.mh.nagasaki-u.ac.jp/soshin/ (accessed on 7 July 2022) and http://www.med.nagasaki-u.ac.jp/cm/ (accessed on 7 July 2022).

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
