# Peer review of "Low-Density Lipoprotein Cholesterol, Structural Atherosclerosis, and Functional Atherosclerosis in Older Japanese"

_nutrients, 2022, doi:10.3390/nu15010183_

Round 1
Reviewer 1 Report
This paper reports studies that authors conducted on an older Japanese population (1,458 participants aged 60 to 79 years) to investigate the relationship between low-density lipoprotein cholesterol (LDLc) and structural atherosclerosis and functional atherosclerosis. They found that LDLc was positively associated with structural atherosclerosis and inversely associated with functional atherosclerosis; and they stated that clinical practice may combine structural and functional assessment of atherosclerosis.
This paper is well-written and data is well-presented. The findings and conclusion are supported by the data with good discussion.
However, given that the relationship between LDLc and atherosclerosis has been well known, the findings lack novelty, although it is surprising that structural atherosclerosis does not correlate with functional atherosclerosis. There are many significant factors (e.g age, drinking status, diabetes, hypertension and usage of lipid-lowering medication) that have confounded the results and conclusion. In addition, it is not clear why the authors did not include an obvious variable –HDL- in their analysis. Further, while the hypothesis indicates a causal relationship between LDLc and atherosclerosis, the findings are merely association not causational.
I also found some sections lack enough information. For example, in Results Section, it is stated that “two hundred thirty-seven participants were diagnosed with structural atherosclerosis and with 494 functional atherosclerosis”. How many participants were diagnosed with both structural and functional atherosclerosis? Are functional and structural atherosclerosis reversely correlated in these patients as well? How many participants have neither? If the purpose is to investigate the association between LDLc and atherosclerosis, why non-atherosclerosis participants were included in the analysis?
Table 2 shows that 556+494 participants had functional atherosclerosis, what are the structural atherosclerosis data?
Finally, to better support the claim that the findings are true in an older population, it needs evidence from the young population. If not possible to include younger patients/participants, I would suggest analyzing the population by age groups (older or younger than 70). I wonder if the results would be different or not when the population is divided by participants’ ages.
Author Response
Reviewer 1
Thank you for valuable comments. According to those valuable comments, we rechecked our present manuscript and revised.
1. However, given that the relationship between LDLc and atherosclerosis has been well known, the findings lack novelty, although it is surprising that structural atherosclerosis does not correlate with functional atherosclerosis.
⇒
Thank you for valuable comments. According to this reviewer’s valuable comment, I rechecked the meaning of structural atherosclerosis and functional atherosclerosis.
In our present study, even LDLc revealed to be significantly positively associated with structural values of atherosclerosis, LDLc also revealed to be inversely associated with functional atherosclerosis. Those are the most important findings of present study. Since the findings that showed positive association between LDLc and structural atherosclerosis is supported by previous studies, the representativeness of present study could be validated to some extent. Therefore, even the significant association between LDLc and structural atherosclerosis is not newly finding, this positive association is important finding to be shown in present study.
Furthermore, as shown in previous studies [Ref1][Ref2][Ref3][Ref4], from the perspective of actively of endothelial repair, there is significant difference between structural atherosclerosis and functional atherosclerosis; aggressive endothelial repair progresses both of structural atherosclerosis and functional atherosclerosis but insufficient endothelial repair furthers functional atherosclerosis but not structural atherosclerosis. And as shown in previous study, CD34-positive cell is necessary to develop structural atherosclerosis [Ref5] while shortage of CD34-positive cell progresses functional atherosclerosis [Ref4]. Since aging is process that decline the productivity of CD34-positive cell, the analysis among elderly could emphasize the influence of shortage of endothelial repair. Therefore, among elderly, functional atherosclerosis could be not correlated with structural atherosclerosis
Therefore, even this reviewer thought it is surprising that structural atherosclerosis does not correlate with functional atherosclerosis, this correlation could be explained by the influence of shortage of endothelial repair that relates to the presence of CD34-positive cell. The findings between structural atherosclerosis and functional atherosclerosis are also important informative knowledge to present study.
[Ref1]
Comment on "Does body height affect vascular function?".
Shimizu Y.Hypertens Res. 2022 Jun;45(6):1091-1092. doi: 10.1038/s41440-022-00887-3.
[Ref2]
Mechanism underlying vascular remodeling in relation to circulating CD34-positive cells among older Japanese men
Yuji Shimizu
Scientific Reports volume 12, Article number: 21823 (2022)
[Ref3]
Tooth Loss and Carotid Intima-Media Thickness in Relation to Functional Atherosclerosis: A Cross-Sectional Study.
Shimizu Y, Yamanashi H, Kitamura M, Miyata J, Nonaka F, Nakamichi S, Saito T, Nagata Y, Maeda T.J Clin Med. 2022 Jul 10;11(14):3993. doi: 10.3390/jcm11143993.
[Ref4]
Cardio-ankle vascular index and circulating CD34-positive cell levels as indicators of endothelial repair activity in older Japanese men.
Shimizu Y, Yamanashi H, Noguchi Y, Koyamatsu J, Nagayoshi M, Kiyoura K, Fukui S, Tamai M, Kawashiri SY, Kondo H, Maeda T.Geriatr Gerontol Int. 2019 Jun;19(6):557-562. doi: 10.1111/ggi.13657.
[Ref5]
Circulating CD34+ cells and active arterial wall thickening among elderly men: A prospective study.
Shimizu Y, Kawashiri SY, Kiyoura K, Koyamatsu J, Fukui S, Tamai M, Nobusue K, Yamanashi H, Nagata Y, Maeda T.Sci Rep. 2020 Mar 13;10(1):4656. doi: 10.1038/s41598-020-61475-4.
To clarify the influence of circulating CD34-positive cell and atherosclerosis (structural and functional), we added following sentences in introduction section.
Low-density lipoprotein cholesterol (LDLc) directly contributes to the development of structural atherosclerosis [3] by activating inflammation [4], even when levels are within the normal range [5]. Structural atherosclerosis requires CD34-positive cells [1], which contribute to endothelial repair [6], while a shortage of CD34-positive cells results in functional atherosclerosis but not structural atherosclerosis [1].
Since LDLc was also reported to increase the proliferation of CD34-positive cells [7], individuals with high LDLc levels should have sufficient circulating CD34-positive cells to induce endothelial repair.
In older people, therefore, LDLc may be positively associated with structural atherosclerosis and inversely associated with functional atherosclerosis as a result of activating endothelial repair.
2. There are many significant factors (e.g age, drinking status, diabetes, hypertension and usage of lipid-lowering medication) that have confounded the results and conclusion. In addition, it is not clear why the authors did not include an obvious variable –HDL- in their analysis.
⇒
Thank you for valuable comment. However, the points that this reviewer pointed out about the confounder in present study is not appropriate. In present study, we made three adjusted models. The first one is adjusted only for sex and age (Model 1). And for Model 2, this model is adjusted for sex, age, weight (underweight, normal weight, overweight), hypertension, drinking status (none, often, daily), smoker (never, former, current), high triglycerides, low high-density lipoprotein cholesterol (HDLc), and diabetes. And for Model 3, adjusted for variables in Model 2 and lipid-lowering medication use. Therefore, we already included HDLc as a confounder. Since HDLc shows no linear association with atherosclerosis, continuous variables are inappropriate to use in this model.
3. Further, while the hypothesis indicates a causal relationship between LDLc and atherosclerosis, the findings are merely association not causational.
⇒
Thank you for valuable comment. According to this reviewer’s valuable comment, I rechecked present hypothesis. As shown in introduction section, we described as “Therefore, LDLc could be positively associated with structural atherosclerosis and inversely associated with functional atherosclerosis”. Then we did not indicate causal relationship between LDLc and atherosclerosis in our present hypothesis.
This is the reason why we did describe as following in discussion section as limitation. “In addition, this was a cross-sectional study and was therefore unable to establish causal relationships.”
Since this study is a scientific study, the hypothesis should be evoked from scientific manner. CD34-positive cell has been revealed to contributing to determine the association between structural atherosclerosis and functional atherosclerosis in cross-sectional study [Ref4]. And LDLc also reported to regulate the production of CD34-positive cell [Ref6].
Then we thought there is a novel mechanism that determine the association between LDLc and structural atherosclerosis and the association between LDLc and functional atherosclerosis. According to those studies, CD34-positive cell might act as a determinant on the association between LDLc and functional atherosclerosis. Therefore, the information of CD34-positive cell should be informative. Even though, we have no data of CD34-positive cell in present study. Then we described as following in limitation section.
CD34-positive cells might have played important roles in our findings, but we have no data on the number of these cells in each individual. Further investigation that includes CD34-positive cell counts is necessary. In addition, this was a cross-sectional study and was therefore unable to establish causal relationships.
[Ref4]
Cardio-ankle vascular index and circulating CD34-positive cell levels as indicators of endothelial repair activity in older Japanese men.
Shimizu Y, Yamanashi H, Noguchi Y, Koyamatsu J, Nagayoshi M, Kiyoura K, Fukui S, Tamai M, Kawashiri SY, Kondo H, Maeda T.Geriatr Gerontol Int. 2019 Jun;19(6):557-562. doi: 10.1111/ggi.13657.
[Ref6]
Cimato TR, Palka BA, Lang JK, Young RF. LDL cholesterol modulates human CD34+ HSPCs through effects on proliferation and the IL-17 G-CSF axis. PLoS One. 2013 Aug 26;8(8):e73861. doi: 10.1371/journal.pone.0073861.
4. I also found some sections lack enough information. For example, in Results Section, it is stated that “two hundred thirty-seven participants were diagnosed with structural atherosclerosis and with 494 functional atherosclerosis”. How many participants were diagnosed with both structural and functional atherosclerosis?
⇒
Thank you for valuable comment. According to this reviewer’s valuable comment, I calculated the number of participants who has both of structural atherosclerosis and functional atherosclerosis. And in present study, 107 participants were diagnosed as having both structural and functional atherosclerosis. Since this number is observed in table 2, and are not main topic of present study we did not add in text.
5. Are functional and structural atherosclerosis reversely correlated in these patients as well? How many participants have neither?
⇒
Thank you for valuable comment. According to this reviewer’s valuable comment, I rechecked the association between structural atherosclerosis and functional atherosclerosis. As shown in table 2, after adjusted for known confounder, the correlation between structural atherosclerosis and functional atherosclerosis became no significant. Decreased circulating CD34-positive cell due to consumption might blame to those associations. To evaluate the influence of developing functional atherosclerosis by status of structural atherosclerosis which relates to reduction in circulating CD34-positive cell due to consumption, we made additional analysis. Then I added following sentences in result section.
3.5. Association between LDLc and functional atherosclerosis according to the status of structural atherosclerosis
In an additional analysis, we repeated the main analysis with stratification by structural atherosclerosis, and found a significant inverse association between LDLc and functional atherosclerosis but only in participants without structural atherosclerosis. The sex- and age-adjusted ORs (95% CIs) of functional atherosclerosis per 1-SD increment of LDLc were 0.99 (0.74, 1.33) for those with structural atherosclerosis (n=207) and 0.78 (0.69, 0.89) for those without it (n=1221), respectively.
And we added following sentences in discussion section.
Aggressive endothelial repair, which is related to the development of structural atherosclerosis, also decreases the number of circulating CD34-positive cells due to consumption [11,27]. This is consistent with the previous finding of a significant inverse association between baseline atherosclerosis and active arterial wall thickening as evaluated by yearly progression of CIMT [11]. In people with structural atherosclerosis, functional atherosclerosis may develop both due to aggressive endothelial repair and to insufficient endothelial repair caused by a shortage of CD34-positive cells due to consumption. Additionally, in individuals without structural atherosclerosis, functional atherosclerosis may arise only as the result of insufficient endothelial repair. These mechanisms suggest that the presence or absence of structural atherosclerosis confounds the association between LDLc and functional atherosclerosis. This could explain why the additional analysis in this study revealed that only individuals without structural atherosclerosis showed a significant inverse association between LDLc and functional atherosclerosis.
Since table 2 shows the number of participants who has structural atherosclerosis by status of functional atherosclerosis and the number of participants who has structural atherosclerosis were 37 for those with CAV<8.0 (n=408), 93 for CAVI: 8.0-8.9 (n=556), and 107 for those with 9.0≤CAVI (n=494). Then (408-37) + (556-107)=834 participants have neither functional and structural atherosclerosis. Since those information are already in table 2, we did not make additional change for those.
[Ref2]
Mechanism underlying vascular remodeling in relation to circulating CD34-positive cells among older Japanese men
Yuji Shimizu
Scientific Reports volume 12, Article number: 21823 (2022)
6. If the purpose is to investigate the association between LDLc and atherosclerosis, why non-atherosclerosis participants were included in the analysis?
⇒
Thank you for valuable comment. According to this reviewer’s valuable comment, I rechecked the aim of present study. To evaluate the association between LDLc and atherosclerosis by calculating odds ratio, participants without atherosclerosis should be include. Furthermore, atherosclerosis is common condition to general population. Then case control study is not necessary. Then we just used the data from general population who participated in annual health check-up. This is the reason why our present study population includes with participants without atherosclerosis.
7. Table 2 shows that 556+494 participants had functional atherosclerosis, what are the structural atherosclerosis data?
⇒
Thank you for valuable comment. According to this reviewer’s valuable comment, I rechecked table 2. Table 2 shows the association between structural atherosclerosis and the status of functional atherosclerosis. Then the number of cases which are shown in Tables indicates the number of structural atherosclerosis. Then the actual number of structural atherosclerosis for each category were 37 for non-functional atherosclerosis, 93 for early functional atherosclerosis, and 107 for functional atherosclerosis.
8. Finally, to better support the claim that the findings are true in an older population, it needs evidence from the young population. If not possible to include younger patients/participants, I would suggest analyzing the population by age groups (older or younger than 70). I wonder if the results would be different or not when the population is divided by participants’ ages.
⇒
Thank you for valuable comment. According to this reviewer’s valuable comment, I rechecked the study plan of present study. We thought CD34-positive cell might takes important role in present main associations. And only study with elderly reported the potential mechanism among CD34-positive cell, functional atherosclerosis, and structural atherosclerosis. Even the productivity of CD34-positive cell could be strongly influenced by aging, no study reported the association among CD34-positive cell, functional atherosclerosis, and structural atherosclerosis in younger population. Therefore, analysis among younger could not possess enough background that support the results. Furthermore, there are no evidence that support the age of 70 is the efficient cut off value for present study. Since present study is evoked from scientific evidence, we should follow scientific evidence. However, with regards with above mentioned reason, we made additional analysis stratified by age groups and found essentially same associations. Among participants aged 60 to 69 years, 1 standard increment of LDLc for functional atherosclerosis and structural atherosclerosis were 1.35 (1.05, 1.74) and 0.83(0.73, 0.997), respectively. Among participants aged 70 to 79 years, 1 standard increment of LDLc for functional atherosclerosis and structural atherosclerosis were 1.15 (0.97, 1.37) and 0.80(0.69, 0.93), respectively. Then we added following sentences in manuscript.
In result section
3.6. Associations between LDLc and both structural and functional atherosclerosis by age group
We also repeated the main analysis with stratification by age group, and found essentially the same associations. The age-adjusted ORs (95% CIs) of structural and functional atherosclerosis per 1-SD increment of LDLc were 1.35 (1.05, 1.74) and 0.83 (0.70, 0.997) for ages 60 to 69 years (n=754), respectively, and 1.15 (0.97, 1.37) and 0.80 (0.69, 0.93) for ages 70 to 79 years (n=704), respectively.
In discussion section
An additional analysis that evaluated the associations between LDLc and both structural and functional atherosclerosis stratified by age group (60 to 69 years and 70 to 79 years) showed essentially the same results. To evaluate the influence of age on the present associations, further research involving younger individuals is necessary.

Reviewer 2 Report
This manuscript by Shimizu et al. describes a cross-sectional study in older Japanese individuals to determine the correlation between LDLc level and structural and functional atherosclerosis. The study has a sound design, analysis, and conclusion. The introduction should be more elaborated to describe the background of the study well. English language editing is required.
Author Response
Reviewer 2
Thank you for valuable comments. According to those valuable comments, we rechecked our present manuscript and revised.
1) This manuscript by Shimizu et al. describes a cross-sectional study in older Japanese individuals to determine the correlation between LDLc level and structural and functional atherosclerosis. The study has a sound design, analysis, and conclusion. The introduction should be more elaborated to describe the background of the study well. English language editing is required.
⇒
Thank you for valuable comments. According to this valuable comment, I added following sentences in Introduction section to clarify the influence of CD34-positive cell on structural atherosclerosis and functional atherosclerosis.
Low-density lipoprotein cholesterol (LDLc) directly contributes to the development of structural atherosclerosis [3] by activating inflammation [4], even when levels are within the normal range [5]. Structural atherosclerosis requires CD34-positive cells [1], which contribute to endothelial repair [6], while a shortage of CD34-positive cells results in functional atherosclerosis but not structural atherosclerosis [1].
Since LDLc was also reported to increase the proliferation of CD34-positive cells [7], individuals with high LDLc levels should have sufficient circulating CD34-positive cells to induce endothelial repair.
In older people, therefore, LDLc may be positively associated with structural atherosclerosis and inversely associated with functional atherosclerosis as a result of activating endothelial repair.
And this manuscript was checked by professional English editing service.
